# Embryonic Signaling Pathways Shape Colorectal Cancer Subtypes: Linking Gut Development to Tumor Biology

**DOI:** 10.3390/pathophysiology32040052

**Published:** 2025-10-01

**Authors:** Kitty P. Toews, Finn Morgan Auld, Terence N. Moyana

**Affiliations:** Diagnostic & Molecular Pathology, Department of Pathology & Laboratory Medicine, The Ottawa Hospital & University of Ottawa, General Campus, 501 Smyth Road, Ottawa, ON K1H 8L6, Canada; ktoews@toh.ca (K.P.T.); fauld@toh.ca (F.M.A.)

**Keywords:** colorectal cancer, right and left colon, mid- and hindgut, embryologic development, signaling pathways, precursor lesions, pathogenesis, consensus molecular subtypes, drug resistance, tumor micro-environment

## Abstract

The morphogenesis of the primordial gut relies on signaling pathways such as Wnt, FGF, Notch, Hedgehog, and Hippo. Reciprocal crosstalk between the endoderm and mesoderm is integrated into the signaling pathways, resulting in craniocaudal patterning. These pathways are also involved in adult intestinal homeostasis including cell proliferation and specification of cell fate. Perturbations in this process can cause growth disturbances manifesting as adenomas, serrated lesions, and cancer. Significant differences have been observed between right and left colon cancers in the hindgut, and between the jejunoileum, appendix, and right colon in the midgut. The question is to what extent the embryology of the mid- and hindgut contributes to differences in the underlying tumor biology. This review examines the precursor lesions and consensus molecular subtypes (CMS) of colorectal cancer (CRC) to highlight the significance of embryology and tumor microenvironment (TME) in CRC. The three main precursor lesions, i.e., adenomas, serrated lesions, and inflammatory bowel disease-associated dysplasia, are linked to the CMS classification, which is based on transcriptomic profiling and clinical features. Both embryologic and micro-environmental underpinnings of the mid- and hindgut contribute to the differences in the tumors arising from them, and they may do so by recapitulating embryonic signaling cascades. This manifests in the range of CRC CMS and histologic cancer subtypes and in tumors that show multidirectional differentiation, the so-called stem cell carcinomas. Emerging evidence shows the limitations of CMS particularly in patients on systemic therapy who develop drug resistance. The focus is thus transitioning from CMS to specific components of the TME.

## 1. Introduction

Colorectal cancer, exemplified by adenocarcinoma, is the most common type of gastrointestinal malignancy in developed countries and regions with similar lifestyles and dietary habits [1,2,3]. It is one of the leading causes of cancer deaths in the world, along with lung, breast, prostate, and pancreas. Despite progress in prevention and treatment, greater than 1 million new cases and greater than half a million CRC deaths are estimated to occur each year worldwide [1,2,3,4]. CRC is a highly heterogeneous disease, with diverse pathophysiologic and molecular features, as well as variable survival outcomes and therapeutic vulnerabilities. This makes it challenging to optimize treatment modalities to reduce morbidity and mortality [5]. The heterogeneity of colorectal cancer is partly attributable to embryologic origin and tumor location, with important distinctions between right- and left-sided tumors recognized as early as 1990 by Bufill [6,7,8]. Since then, there have been major developments in CRC biology, including genomics, transcriptomics and consensus molecular subtypes (CMS). This review takes these developments into account, comparing similarities between embryonic signaling pathways which are coordinated and physiologic versus CRC which is characterized by aberrant and uncontrolled growth due to genomic alterations. Since CRC patients on systemic therapy commonly develop drug resistance, the review also describes the limitations of the CMS classification and how studies are transitioning to focus on the tumor micro-environment.

## 2. Embryologic Development

The development of the primordial gut is a complex process that relies on many signaling mechanisms. Of the three primary germ layers arising post-gastrulation, the endoderm forms the epithelial layer of the mucosa; the mesoderm makes the muscular layer, lamina propria, connective tissue, and blood vessels; and the ectoderm creates the enteric nervous system (Figure 1) [9,10,11,12]. From the GI embryologic anlage, intestinal regional identity is established along the craniocaudal axis resulting in the formation of the foregut, midgut, and hindgut. In the midgut, the ileocecal region is formed from a complex interplay of intestinal physiologic herniation into the umbilical cord, rotation and intussusception of the ileum into the cecum while the appendix arises from the cecal diverticulum [9,10].

In the small intestine, Hedgehog signals from Shh and Ihh drive mesenchymal development, which then expresses BMPs and PDGFR-α to promote villus formation, while Sox9 simultaneously suppresses villification in the intervillous regions [13,14] (Figure 1). A tightly regulated crypt-villus axis is thus established that enables the proliferation and differentiation of new epithelial cells every three to seven days throughout life [10,15]. Intestinal homeostasis and regenerative output are maintained by multipotent stem cells which reside in a stem cell niche at the crypt base, and the process is controlled by balancing signaling involving Notch, Wnt, and BMP pathways [10,13,14,15,16] (Table 1).

More distally, several signaling pathways promote posterior axis patterning, including *WNT* and *FGF*. Studies suggest that the concentration and duration of exposure to signaling pathway activation (i.e., signaling gradients) influence regional intestinal identity. Therefore, prolonged activation of *FGF* and *WNT* signaling patterns the intestine into increasingly distal tissue. Signaling input also involves other pathways including caudal type homeobox (CDX) and the highly complex Hippo pathway, whose components include transcriptional co-activators Yes-associated protein (YAP) and Transcriptional coactivator with PDZ-binding motif (TAZ) [11,12,13,14,15,16].

To illustrate the importance of signaling, studies on organoids have shown that ligand stimulation of mesenchymal-epithelial interactions can switch the cell fates of the foregut and hindgut identities, thus highlighting the mechanisms of lineage plasticity [17]. Furthermore, following intestinal cell damage, TGF-β can reprogram enterocytes, goblet cells, and paneth cells into revival stem cells that then reconstitute leucine-rich repeat-containing G protein-coupled receptor 5 (Lgr^5+^) intestinal stem cells to kick-start regeneration [18].

Together, these findings highlight a complex network in which transcriptional regulators mediate reciprocal crosstalk between the endoderm and mesoderm, establishing craniocaudal patterning and domain identity of the GI tract. Notably, the same signaling pathways and epigenetic mechanisms also govern adult gut homeostasis, regulating processes such as cell proliferation, polarity, and fate specification [19,20]. Perturbations in this process can cause disturbances in growth, which can manifest as adenomas, serrated lesions, and cancer. Therefore, the embryologic underpinnings of the mid- and hindgut likely contribute to differences in the underlying tumor biology.

## 3. Right-Versus Left-Sided Cancers

The right colon (cecum, ascending and proximal two-thirds of the transverse colon) is derived from the midgut, while the left (distal one-third of the transverse colon, descending and sigmoid colon, and rectum) is derived from the hindgut (Figure 2) [6,7,8]. 

This distinction is reflected in the blood supply, with the right colon supplied by the superior mesenteric artery and the left colon by the inferior mesenteric artery. The overall incidence of right-sided cancers is lower than left-sided ones. However, the proportion of right-sided tumors has steadily increased over the years. Right-sided cancers are more prevalent in women and in older individuals. Since the right colon has a wide diameter, the cancers tend to be large exophytic masses (Figure 3A), presenting with subtle features such as anemia and weight loss. As a result, these cancers are often diagnosed at more advanced stages [6,7,8]. This may be partly explained by the higher rate of incomplete colonoscopies on the right side and the increased prevalence of sessile serrated lesions (SSLs), whose flat morphology and indistinct borders make them more difficult to detect [21,22]. Unlike SSLs, conventional tubular and tubulovillous adenomas are more uniformly distributed throughout the colon [20].

Left-sided tumors tend to present with obstructive features since they occupy a larger diameter of the colon lumen (Figure 3B). Due to their distal location, hematochezia is a relatively common feature. The histologic repertoire is also different with more mucinous, signet-ring cell and medullary subtypes occurring on the right side [20]. The incidence of peritoneal carcinomatosis is higher with right-sided cancers while that of hepatic and pulmonary metastases is higher on the left. The prognosis for left-sided tumors has been improving over the past 50 years, while that for the right remains more guarded [8,23,24,25,26]. Importantly, within the context of left-sided colon cancers, rectal cancers are often regarded as a separate entity because localized stage 2–3 rectal cancers have a different treatment paradigm. However, systemic therapies for metastatic rectal and left colon cancers are essentially the same.

These right–left differences may partly reflect their distinct embryologic origins. Given that the midgut also gives rise to most of the small intestine and the appendix, comparing cancers from these sites with those of the large bowel may provide additional insights into disease biology.

## 4. Comparisons of Epithelial Cancers of the Midgut

The midgut begins at the post-ampullary duodenum. The D3/D4 segment is relatively short, and the majority of the midgut’s small bowel component consists of the jejunoileum (JI). The JI is quite extensive, making up approximately 90% of the surface area of the small and large bowel combined, yet it accounts for <5% of all GI cancers, compared to CRC which makes up >40% [26,27,28,29,30]. JI epithelial cancers are rare, and the main tumor types are neuroendocrine tumors (NETs) and adenocarcinomas, in that order. This contrasts sharply with the right colon, where adenocarcinomas predominate over NETs. [30,31].

The rarity of JI epithelial cancers remains poorly understood, but several factors have been proposed. These include the following: (i) the relatively fluid nature of intestinal contents, which causes less mucosal irritation compared with the more solid fecal matter of the large bowel; (ii) the absence of a storage function in the small intestine, limiting prolonged exposure to potentially harmful ingested elements, unlike the longer transit time in the large intestine; (iii) the generation of fewer reactive oxygen species, which reduces DNA damage and, consequently, cancer risk; (iv) the relative richness of the colonic microbiome and its associated biofilms, some of which contribute to CRC development [23]; (v) differences in bile acid and salt composition between the small and large bowel; and (vi) the small intestine contains abundant lymphoid tissue and serves as a major site of immunosurveillance, where mucosal dendritic cells sample antigens from food and microbiota and modulate CD4+ and CD8+ lymphocyte activity. Notwithstanding the differences between the JI and right colon, the appendix, which is located between them, has its own distinct oncologic profile. The most common cancer is the NET, which exhibits a very different biology from that in the JI [32]. For example, >90% of appendiceal NETs are cured by simple appendectomy, whereas 50% of JI have distant metastases [33]. Other fairly common appendiceal neoplasms are goblet cell carcinomas and low-grade appendiceal mucinous neoplasms (LAMNs), which are rarely encountered in the JI [34]. Extending the comparisons to the large bowel, right colonic cancers are dominated by adenocarcinomas with a relatively much smaller proportion of NETs and, rarely, goblet cell carcinomas and low-grade mucinous (appendiceal-type) neoplasms [34]. The reasons for these site-specific differences are not entirely clear, but the embryologic development of the gut may provide some explanation.

## 5. Precursor Lesions for CRC

The principal precursor lesions of CRC include adenomas (Figure 4A,B), sessile serrated lesions, and inflammatory bowel disease-associated dysplasia (Table 2).

### 5.1. Conventional Adenomas

These are the most common of adenomas and include tubular, tubulovillous, and villous adenomas.

The formation of adenomas is associated with a number of genetic changes, characteristically the chromosomal instability pathway involving *APC*, *KRAS*, *TP53*, and *SMAD4* (Figure 5). Inactivating mutations in *APC* occur in 60–80% of adenomas, and appear early in the sequence [20,35,36,37]. These alterations often result in truncation of the *APC* protein which prevents phosphorylation of β-catenin, thereby reducing its degradation, which then accumulates. Ultimately, β-catenin translocates to the nucleus, driving *LEF/TCF*-mediated gene transcription. Activation of *Wnt* pathways by these mechanisms does not, by itself, cause cancer, though it is important in initiating the carcinogenic process.

The same mechanism is the main driver leading to the formation of the multiple adenomas found in familial adenomatous polyposis syndrome (*FAP*) (Figure 6) [20,35,36,37]. The progressive enlargement of adenomas is accompanied by further molecular abnormalities such as mutations involving *KRAS*, *SMAD4*, *PTEN*, *PIK3CA*, and *TP53*. As the adenoma enlarges, there is concomitant stimulation of additional blood supply (neo-angiogenesis) and stroma to support its growth.

### 5.2. Serrated Pathway

The first descriptions of sessile serrated lesions (SSLs) appeared in the 1990s but their clinical significance was not initially well appreciated (Figure 7) [21].

It is now known that up to 30% of sporadic CRCs develop through the serrated neoplasia pathway with the main precursors being SSLs and traditional serrated adenomas (Figure 8) [21,22,38,39]. In contrast to the conventional adenoma pathway, serrated precursor lesions (mucin-vacuolated hyperplastic polyps and SSLs) rarely present with *APC* mutations [21,39]. Instead, they tend to initially manifest *BRAF/KRAS* mutations and hypermethylation, which leads to the activation of the *MAPK* cascade, causing the disruption of crypt cell proliferation and differentiation. 

*BRAF* mutations occur in ≥70% of SSLs compared with only 6% in non-serrated lesions, which suggests an initiating role (Figure 9). In addition to *BRAF* mutations, the serrated pathway is strongly associated with hypermethylation, as shown by microsatellite instability (MSI) and the CpG island methylator phenotype (CIMP); among these, CIMP is the dominant mechanism [38,40]. CIMP plays a significant role in driving the progression of benign lesions to malignant serrated CRC by causing the inactivation of several tumor suppressor genes, including *TP53* and *CDKN2A* [38]. In contrast, goblet cell-rich hyperplastic polyps and traditional serrated adenomas, which are more common in the left colon, are characterized by *KRAS* mutations and low CIMP. Taking the serrated neoplasia pathway as a whole, *KRAS* mutations are less frequent (<10%) and appear to be mutually exclusive with respect to *BRAF* [40].

### 5.3. Inflammatory Bowel Disease-Associated Dysplasia

Individuals with IBD (Figure 10A,B) have a two to three times higher risk of developing CRC than those who do not. Overall, the percentage of CRC associated with IBD is low at ≤1% [41]. The primary driver of colitis-associated CRC is a cumulative inflammatory burden which causes increased oxidative stress and dysbiosis of intestinal microbiota [42]. This results in expansion of pro-tumorigenic clones that replace wild-type colorectal epithelium [42,43]. Due to the field cancerization effect, where widespread mucosal injury and molecular alterations predispose large areas of epithelium to neoplastic change, the dysplasia in IBD is often multifocal and can be synchronous or metachronous [43,44]. The genetic makeup of colitis-associated CRC is somewhat similar to sporadic CRC, with some of the most recurrent gene mutations being *TP53*, *KRAS*, and *SMAD4*. However, there are differences in the sequence and timing of alterations in the carcinogenesis process, e.g., *TP53* alterations are an early and highly recurrent event [44,45]. Furthermore, *APC* mutations, which are observed in the majority of sporadic microsatellite stable (MSS) CRC cases, are detected in much smaller proportions in IBD dysplasia/carcinoma. This suggests the presence of an alternative mechanism of tumor initiation in IBD [43,44].

## 6. Histologic Subtypes of CRC

The majority of CRCs are diagnosed as adenocarcinoma not otherwise specified (NOS) (Figure 11A), that is, they are composed of mundane malignant glands without specific distinguishing features [34,46]. However, there is a smaller proportion of cases where histopathologic subtypes can be defined, with specific clinicopathologic characteristics (Table 3).

These subtypes include medullary, mucinous, signet-ring cell, serrated, micropapillary, adenoma-like, adenosquamous, sarcomatoid, undifferentiated, small/large cell neuroendocrine, and invasive-stratified mucinous carcinomas [34,46,47,48]. Medullary carcinoma (Figure 11B) can occur either sporadically or in association with Lynch syndrome [34,46,47]. It is associated with microsatellite instability (MSI), a solid growth pattern, and relatively good prognosis. By contrast, signet-ring cell carcinoma tends to be highly infiltrative and has adverse prognostic significance independent of stage.

Mucinous adenocarcinomas (Figure 12A) appear to be a heterogenous group with pathogenetic associations variously linked to adenocarcinoma NOS, signet-ring cell and invasive stratified mucinous carcinomas. The micropapillary subtype (Figure 12B) is characterized by small, tight clusters of tumor cells in cleft-like spaces and relatively early lymphovascular invasion. Serrated adenocarcinomas are thought to be related to serrated adenomas and have a worse prognosis than adenocarcinoma NOS. Neuroendocrine carcinomas are characterized by positivity for neuroendocrine markers and generally have poor outcomes [34,46,47,48]. Lastly, there are CRCs that show bidirectional or even multidirectional differentiation (stem-cell carcinomas) [49,50,51,52,53,54]. These histologic classifications and biologic features highlight the heterogeneity of CRC, and this in turn is a clinical challenge for the selection and optimization of treatment modalities. Thus, more recently, there has been a shift towards transcriptome-based classifications, which resulted in the identification of four consensus molecular subtypes (CMSs).

## 7. Consensus Molecular Subtypes (CMS)

The consensus molecular subtype (CMS) classification for CRC, based on bulk transcriptomic profiling, is a collection of entities with characteristic histologic, genomic, molecular, and clinical features [25,55,56,57]. While it is mainly slanted towards capturing the intrinsic biomolecular heterogeneity of CRC, it also provides possibilities for estimating prognostic and predictive values [57]. The four recognized subtypes are designated CMS1–4 (Table 4) [51,55,56,58].

CMS1 [23,55] (immunogenic subtype) is predominantly composed of right-sided CRCs and typically shows a strong association with SSLs, high rates of MSI-H, distinct methylation patterns (CIMP) and *BRAF* mutations. From a prognostic perspective, CMS1 has a significantly better prognosis in early stages (stages 2–3) and better survival rate than CMS4. It should be noted that this advantage is reversed in stage 4 disease, and CMS1 stage 4 tumors have historically been associated with the worst survival outcomes. This paradoxical shift is attributed to several factors, including immune evasion, stromal reorganization, molecular cofactors, and therapy resistance [25]. However, given emerging evidence for the efficacy of immune checkpoint inhibitors in the treatment of MSI-high CRC, the poor prognosis associated with CMS1 stage 4 disease could soon change quite significantly [23].

CMS2 (canonical subtype) has epithelial characteristics, chromosomal instability, high somatic copy number alterations (SCNA), and *WNT*, *MYC*, and *SRC/EGFR* pathway activation. [23,25]. CMS2 tumors are predominantly left-sided and have the most favorable prognosis in both local and metastatic disease.

CMS3 (metabolic subtype) has marked activation of multiple metabolic pathways [12], low SCNA, low CIMP, mixed MSI status, and a high rate of *KRAS* mutations [25,55]. It also has epithelial features but less chromosomal instability (CIN) than CMS2. In local disease, CMS2 and CMS3 predict benefit from adjuvant chemotherapy, in contrast to CMS1 and CMS4.

CMS4 (mesenchymal subtype) is enriched for signatures of epithelial–mesenchymal transition (EMT), the pro-EMT *TGF-β* and angiogenic pathways. CMS4 is further distinguished by elevated levels of cancer-associated fibroblasts and by increased activity of YAP and TAZ, the principal effectors of the Hippo pathway [55,57]. These tumors may align more closely with the early-dissemination model, in which tumor cells spread very early in the disease process, as opposed to the classical linear-progression model, in which metastasis occurs only after stepwise genetic evolution to advanced disease. They are also strongly associated with tumor budding [55,59]. CMS4 has the worst overall survival (OS) among all stages combined [23].

### 7.1. Prevalence of CMSs

It is estimated that, overall, the proportions of CMS 1–4 are 15%, 35%, 15%, and 25%, respectively, with 10% being unclassified/mixed (Figure 13) [55,60]. Although studies provide estimates of the prevalence of CMS types, this can vary based on a number of factors, particularly disease stage. For example, the proportion of CMS4 cases ranges from 10% in stage 1 to 40% in stage 4 disease. Furthermore, CMS status may change with disease progression, for example, a tumour may evolve from CMS3 to CMS4 in the course of the disease process [55].

### 7.2. Limitations of CMS

#### 7.2.1. Overlap

CRC frequently belongs to several CMS which creates some degree of overlap [55]. Although genotypic features such as *KRAS* and *BRAF* mutation status are enriched in some subtypes (e.g., *BRAF* in CMS1; *KRAS* in CMS3), their presence or absence do not specifically define any subtype, demonstrating the limited value of genotype in defining broader CRC biology. Similarly, there is heterogeneity among right-sided CRCs, such that even though right-sided CRCs are more often categorized as CMS1 and CMS3, all 4 CMSs are represented among right-sided CRCs [23]. Intratumoral heterogeneity to some extent restricts the clinical utility of CMS [57,58]. Some investigators now regard colon tumors as weighted combinations of different CMS subsets instead of ascribing a unique CMS class. This is more in line with the concept of the colorectal continuum model which espouses that molecular alterations occur gradually along the colon [61].

#### 7.2.2. Treatment and Drug Resistance

With treatment, new signaling pathways can be formed, thus altering the CMS profile and inducing therapy resistance. For instance, studies of resistance to WNT-targeted therapy have shown that intestinal tumor cells with common cancer-associated mutations can adapt to evade WNT inhibition and rapidly evolve into a completely WNT-independent state. *WNT* independence is initiated by canonical *TGFβ* signaling, which drives *YAP/TAZ*-dependent transcriptional reprogramming and lineage reversion from adult intestinal epithelium to the fetal state [62].

#### 7.2.3. Cost/Benefit Analysis

Extensive genomic and transcriptomic profiling of tumors is time-consuming and expensive, and its effectiveness requires a cost/benefit analysis. Thus, currently, it tends to be limited to larger centres.

Overall, while the CMS classification has furthered our understanding of CRC, there are limitations to its utility, particularly in the management of patients with advanced or metastatic CRC. Since both chemo- and radiotherapy can be associated with considerable side effects and limited efficacy, combined approaches with immunotherapy and targeted therapies have been beneficial. Key driver gene mutations that have been targeted include *KRAS*, *BRAF*, *EGFR*, *VEGF*, and *MEK*. However, a progressive accumulation of gene mutations and immune escape often lead to drug resistance and treatment failure. Studies suggest that the tumor micro-environment (TME) plays a very important role in these processes [63]. Therefore, there is currently a transition from emphasizing CMSs to focusing on specific components of the TME [63].

## 8. The Tumor Micro-Environment

The TME is a complex network of heterogenous cells and non-cellular components predominantly composed of cancer-associated fibroblasts, immune cells, endothelial cells, and the extracellular matrix (Figure 14). The immune cells include tumor-associated macrophages, tumor-infiltrating lymphocytes, tumor-associated neutrophils, and myeloid-derived suppressor cells. These cells play an important role in tumor proliferation, angiogenesis, and immune escape by interacting with cancer cells, and secreting a variety of cytokines, chemokines, and growth factors [63]. The major components are as itemized below:

Cancer-associated fibroblasts (CAFs) are a varied group and include myofibroblasts, antigen-presenting cells, and vascular-associated CAFs. They secrete a series of soluble factors that activate various signaling pathways including the *β-catenin* pathway, *Hippo-YAP1*, and *BMP*, and in this way play a key role in the remodeling of the extracellular matrix, tumor growth, immune evasion, and drug resistance.

Tumor-associated macrophages are an integral part of the TME. By a process called macrophage polarization, they differentiate into two distinct functional types, namely, M1 and M2. The M1 type is responsible for presenting tumor-specific antigens and releasing pro-inflammatory cytokines such as IL-1, IL-6, and TNF-α to augment the anti-tumor immune response. On the other hand, M2 type macrophages secrete anti-inflammatory factors such as IL-10 and TGF-β which mainly mediate anti-inflammatory reactions and Th2-type immune responses. However, it should be recognized that M1 and M2 polarization is a dynamic process in which their ratios can change in response to the TME. As such, from the viewpoint of drug development, this provides an opportunity for the targeted reprogramming of M1 and M2 [64,65].

Tumor-infiltrating lymphocytes (TILs) associated with the TME have a higher level of specific immunological reactivity against cancer cells than ordinary non-infiltrating lymphocytes [66]. They are a diverse group mainly comprising B-cells, T-cells, and natural killer cells with various phenotypic and functional properties. The significance of TILs is highlighted by colorectal medullary carcinoma, a tumor that is characterized by deficient DNA mismatch repair/high microsatellite instability and extensive lymphocyte infiltration (Figure 7B) [67]. Immune checkpoint inhibitors (ICIs) such as pembrolizumab, nivolumab, dostarlimab, and ipilimumab are much more effective in these tumors compared to microsatellite stable CRCs [63,67]. They enhance anti-tumor effects by blocking immune checkpoint molecules, reactivating the effect of T-cells against tumors, and restoring the immune micro-environment. Unfortunately, since most CRCs (80–90%) are microsatellite stable, this limits the utility of ICIs in clinical practice. Hence, there are ongoing studies to enhance the immunogenicity of microsatellite stable tumors [68,69,70,71]. These studies include but are not limited to the following: (a) artificially introducing neoantigens using β-N-methylamino-L-alanine, which can break immune tolerance without disrupting systemic immune balance; (b) integrating ferroptosis inducers with immunotherapeutic strategies; (c) exploring potential therapeutic strategies that target oxidized low-density lipoprotein; and (d) using a new generation of nanoformulations such as taurine-derived carbon dots which can reprogram the TME by stimulating antitumor immunity [68,69,70,71].

The gut microbiome is a complex ecosystem of microorganisms including bacteria, fungi, viruses, and parasites that interact with each other to form an ecological niche that is important in host physiology and homeostasis [63,72,73]. Perturbations in this process, or *dysbiosis*, can lead to an imbalance that favors pathogenic microorganisms over beneficial ones, e.g., an increase in certain strains of *Clostridium*, *Fusobacterium*, *Bacteroides*, and *Escherichia coli* can lead to a reduction in the prevalence of others such as *Bifidobacterium*, *Lactobacillus*, and *Faecalibacterium* [72,74]. The resultant dysbiosis can contribute to CRC initiation and/or progression through carcinogenic metabolite production, inflammation induction, DNA damage, and oncogenic signaling activation. The altered microbiome can also reduce the efficacy of chemotherapy and immunotherapy. This in turn has spurred efforts to optimize the microbiome to promote a favorable response to therapy [63,75].

## 9. Conclusions

The embryology of the GI tract is a complex process requiring coordinated cross-talk between endoderm and mesoderm to produce patterning in a craniocaudal sequence. This is carefully controlled by signaling cascades to maintain intestinal homeostasis. The similarity of the pathways and stem cells for both intestinal embryogenesis and carcinogenesis is noteworthy. The difference, however, is that in embryogenesis the process is coordinated and homeostatic, whereas in carcinogenesis it is dysregulated by mutations. At the same time, the site-specific differences in the incidence and repertoire of cancers of the JI, appendix, and right colon (all constituents of the midgut) suggest that other factors may be involved in tumorigenesis. The regional differences can also be extended to the hindgut since certain tumors (e.g., medullary, mucinous, and signet-ring cell) are more common on the right, whereas adenocarcinoma NOS is more prevalent on the left. This could be related to regional differences in the micro-environment and pathophysiology, e.g., microbiome, lymphoid tissue, bile acid metabolites, oxyradicals, and transit times. In conclusion, both embryologic and micro-environmental underpinnings of the mid- and hindgut contribute to the differences in the tumors arising from them; tumors may be reactivating and recapitulating embryonic signaling pathways, albeit in aberrant ways through mutations. This is manifested not only in the range of CRC CMSs and histologic cancer subtypes but also in tumors that show multidirectional differentiation, the so-called stem cell carcinomas [49,50,51,52,53,54]. Emerging evidence shows limitations of CMSs, particularly in patients on systemic therapy who develop drug resistance and treatment failure as a result of accumulating gene mutations and immune escape. Thus, there is currently a transition from CMSs to focusing on specific components of the TME.

## Figures and Tables

**Figure 1 pathophysiology-32-00052-f001:**
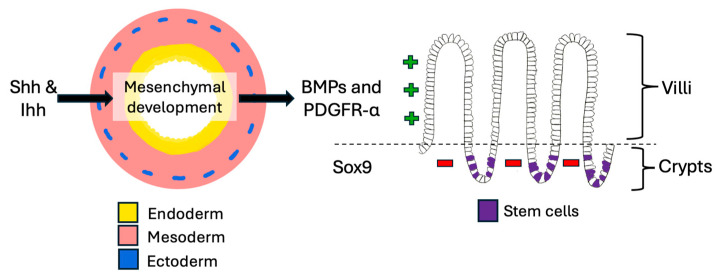
Small intestinal development. The crosstalk between endoderm and mesoderm is very important in the embryologic development of the gut and continues throughout life. It is mediated by various signaling pathways which also contribute to stem cell maintenance.

**Figure 2 pathophysiology-32-00052-f002:**
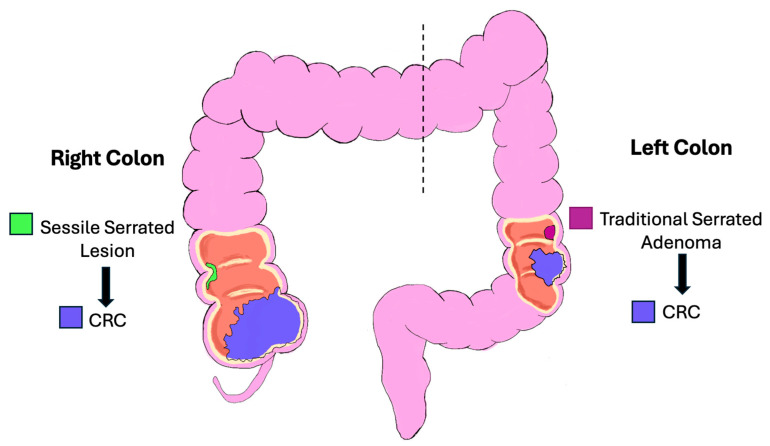
Schematic diagram of the right and left colon. Colorectal cancer can be viewed as originating from distinct regions, e.g., right and left colon and rectum, each with characteristic features based on embryology, clinicopathologic, and molecular features.

**Figure 3 pathophysiology-32-00052-f003:**
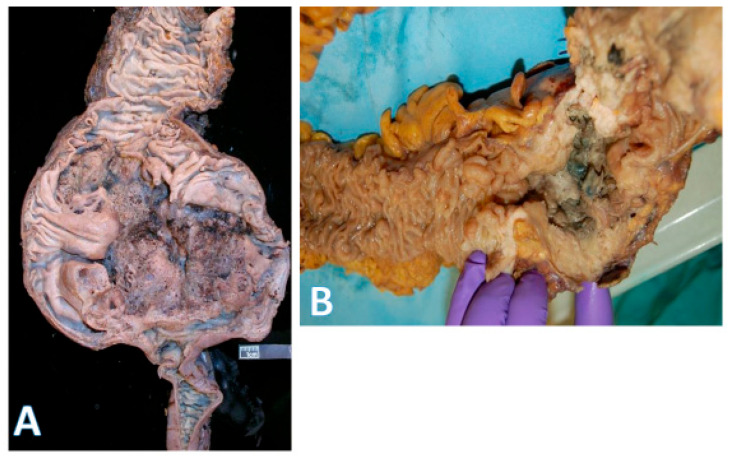
Gross morphology of colorectal cancer. (**A**) on the left shows a very large (14 × 9 cm) cecal adenocarcinoma. The terminal ileum is at the bottom and ascending colon at the top; (horizontal white bar represents 2.6 cm). (**B**) on the right shows a rectosigmoid circumferential (4 × 4 cm) adenocarcinoma (area with gloved fingers) that caused near obstruction of bowel. The proximal aspect is in the upper right field, and the rectal side is on the left.

**Figure 4 pathophysiology-32-00052-f004:**
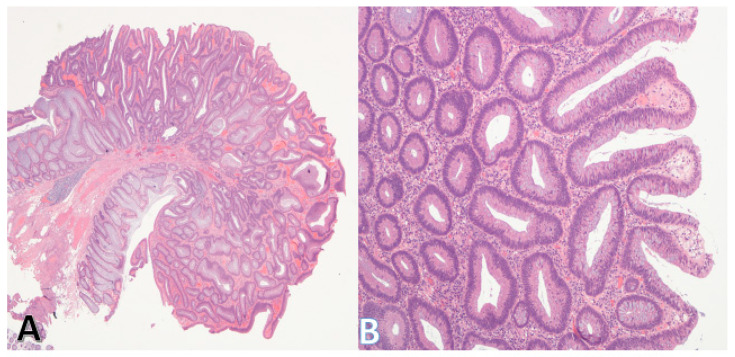
Morphology of the tubular adenoma. (**A**) (on left) shows a low-power view of a tubular adenoma. The stalk and base (bottom left) are clear. (Original magnification ×20). (**B**) shows a higher power view of the adenoma. The lining epithelium is dysplastic, but the individual glands are discrete consistent with low-grade dysplasia. (Original magnification ×100).

**Figure 5 pathophysiology-32-00052-f005:**
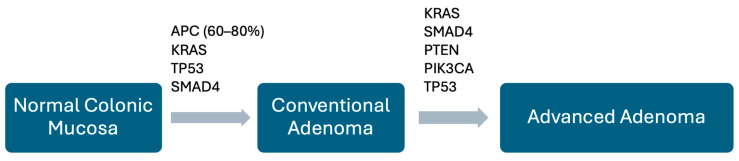
Schematic diagram of change from normal mucosa to adenoma. A number of genomic alterations drive the change from normal colorectal mucosa to adenoma. The key mutations are shown here, though their timing/sequence may vary.

**Figure 6 pathophysiology-32-00052-f006:**
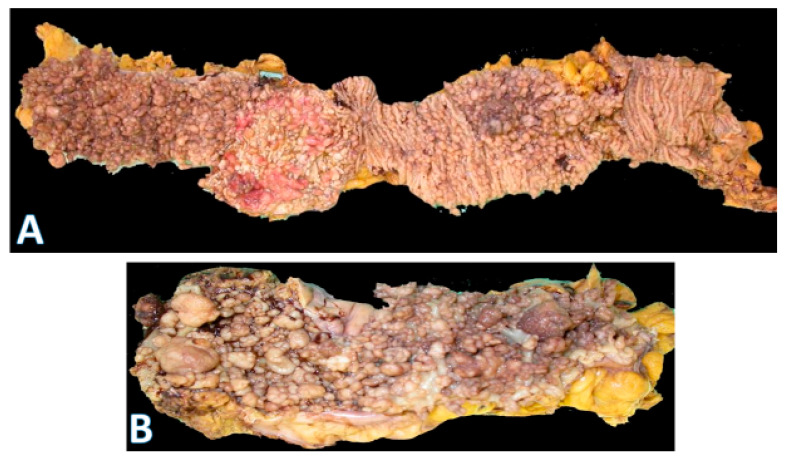
Gross morphology of the colorectum in familial polyposis coli. (**A**) is part of a total colectomy showing hundreds of polyps throughout the length of the large bowel. The majority are tubular adenomas; (horizontal white bar represents 2.0 cm). (**B**) (bottom) is a close-up image showing that, although most of the polyps are small, there are several larger ones (on the left side) and near the end on the right.

**Figure 7 pathophysiology-32-00052-f007:**
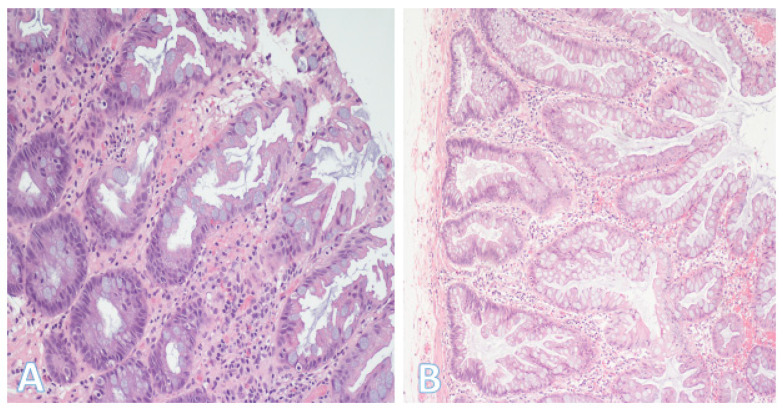
Hyperplastic polyp and sessile serrated lesion. (**A**) On the left is a hyperplastic polyp, one of the most common colorectal polyps. It typically has a sawtoothed architecture, with the serrations generally limited to the upper half of the glands. (Original magnification ×200). (**B**) A sessile serrated lesion. The glands also have a serrated appearance but extend to the basal zone with formation of bulbous and/or horizontal outpouchings. In the figures, the lumen is on the right side. (Original magnification ×200).

**Figure 8 pathophysiology-32-00052-f008:**
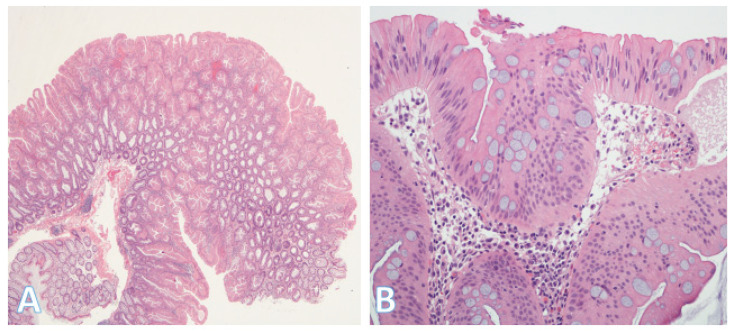
Traditional serrated adenoma. (**A**) On left side is a traditional serrated adenoma with a serrated glandular architecture which is apparent even at this low-power view. (Original magnification ×40). (**B**) A higher-power view showing colonocytes with stratification, eosinophilic cytoplasm, and dysplastic elongated (pencillate) nuclei. (Original magnification ×400).

**Figure 9 pathophysiology-32-00052-f009:**

Schematic representation of the serrated pathway. Diagram showing key genetic alterations and pathways involved in the serrated pathway.

**Figure 10 pathophysiology-32-00052-f010:**
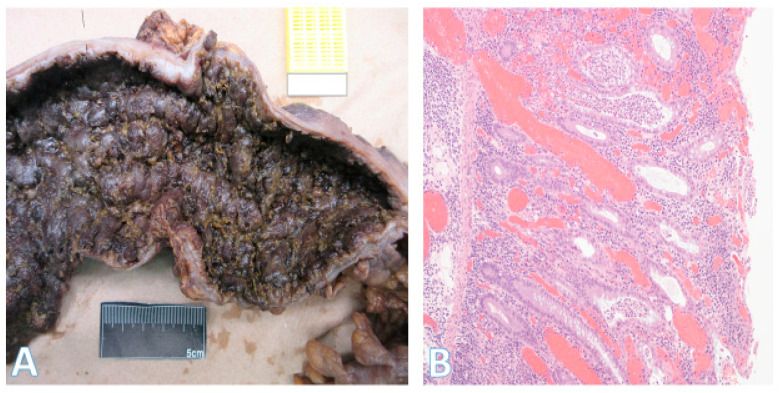
Morphology of severe ulcerative colitis. (**A**) This is part of a total colectomy specimen that was resected for ulcerative colitis with fulminant colitis and failure of medical therapy; (**B**) Section from the same specimen showing severe chronic active inflammation with crypt abscesses and marked mucosal congestion (colon lumen aspect is on the right). (Original magnification ×100).

**Figure 11 pathophysiology-32-00052-f011:**
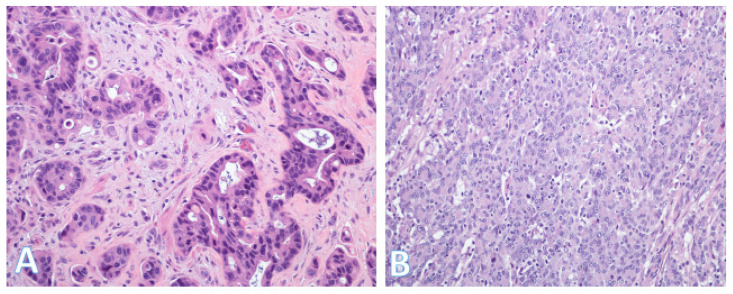
Histologic subtypes of colorectal cancer. (**A**) This shows colorectal adenocarcinoma (NOS) characterized by banal glands in a desmoplastic stroma. Some glands have well-formed lumina consistent with a moderately differentiated (grade 2) adenocarcinoma; (Original magnification ×200). (**B**) This is a medullary carcinoma. In contrast to (**A**), there is generally no glandular formation. Instead, there is a diffuse proliferation of round or polygonal cells with fairly large vesicular nuclei. The cells with the smaller denser nuclei are lymphocytes, which are characteristic of medullary carcinoma. (Original magnification ×200).

**Figure 12 pathophysiology-32-00052-f012:**
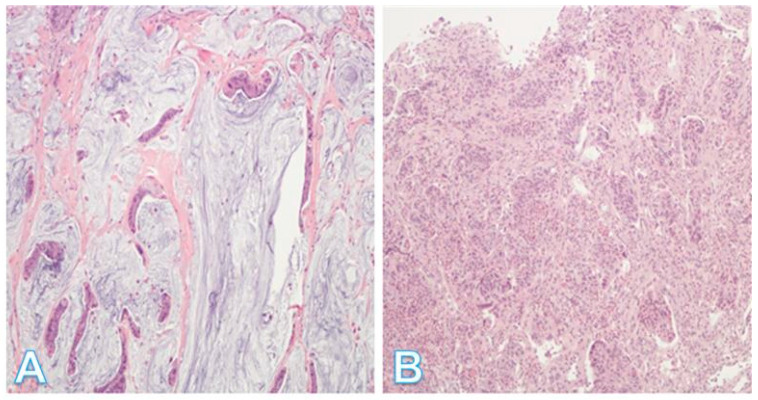
Additional histologic subtypes of colorectal cancer. (**A**) This shows a mucinous adenocarcinoma. Groups of malignant cells are dispersed within the mucin, which forms >50% of the tumor; (Original magnification ×200). (**B**) This shows a micropapillary adenocarcinoma. This is a less common type of tumor showing small, rounded nests with retraction artifact. The tumor has a propensity for lymphovascular invasion. (Original magnification ×100).

**Figure 13 pathophysiology-32-00052-f013:**
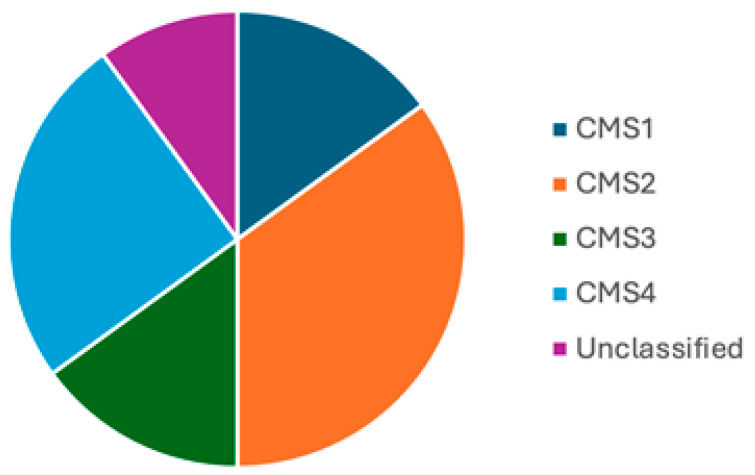
Prevalence of CMS subtypes. Pie chart showing the prevalence of CMS subtypes though this can vary depending on disease stage.

**Figure 14 pathophysiology-32-00052-f014:**
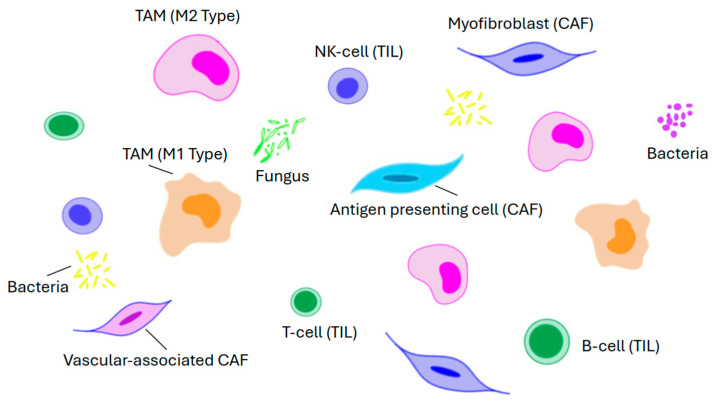
Diagram showing key elements that make up the tumor micro-environment.

**Table 1 pathophysiology-32-00052-t001:** Embryologic signaling pathways in gut development and links to colorectal carcinoma (CRC). Overview of signaling pathways that influence gut development and homeostasis, noting their potential relevance to tumor biology when dysregulated or reactivated.

Process/Region	Key Signals and Regulators	General Functional Role	General Clinical/Tumor Biology Relevance
Germ layer contributions	Endoderm, mesoderm, ectoderm	Establish formation of gut epithelium, mesenchyme, and enteric nervous system	Early developmental programs may be reactivated in cancer biology
Regional identity (foregut, midgut, hindgut)	WNT, FGF, CDX transcription factors	Pattern craniocaudal axis and segmental identity	Regional differences in signaling may contribute to site-specific CRC biology
Ileocecal/appendix formation	Morphogenetic processes (herniation, rotation, cecal budding)	Shape midgut structures including cecum and appendix	Developmental variations may underlie anatomic and biological differences in tumors
Crypt–villus axis formation	Hedgehog, BMP, PDGFR-α, Sox9	Guide villus emergence and crypt–villus organization	Maintenance of epithelial turnover; dysregulation associated with polyp formation
Crypt stem cell niche	Notch, WNT, BMP	Balance self-renewal and differentiation of intestinal stem cells	Disturbances predispose to adenomas and other precursor lesions
Posterior axis patterning	WNT, FGF gradients	Promote distal intestinal identity	May influence molecular subtypes of distal CRC
Hippo pathway	YAP, TAZ	Regulate growth, polarity, and stem cell behavior	Overactivation associated with aggressive CRC features
Lineage plasticity and regeneration	Mesenchymal–epithelial signals; TGF-β; stem cell factors	Allow cell fate flexibility and epithelial regeneration after injury	Reactivation of plasticity pathways implicated in progression and therapy resistance

Abbreviations: CRC, colorectal carcinoma; FGF, fibroblast growth factor; WNT, wingless/integrated; CDX, caudal type homeobox; BMP, bone morphogenetic protein; PDGFR-α, platelet-derived growth factor receptor alpha; Sox9, SRY-box transcription factor 9; Notch, Notch signaling pathway; Hippo, Hippo signaling pathway; YAP, Yes-associated protein; TAZ, transcriptional co-activator with PDZ-binding motif; TGF-β, transforming growth factor beta.

**Table 2 pathophysiology-32-00052-t002:** Precursor lesions of colorectal carcinoma (CRC). Comparison of major precursor lesions (adenomas, serrated lesions, and IBD-associated dysplasia), showing morphologic patterns, char-acteristic genetic alterations, and clinical risk associations.

Precursor Lesion	Morphology/Pathology	Key Molecular Events	Clinical Notes/Risk
Adenomas (conventional pathway)	Tubular, tubulovillous, or villous architecture; progressive dysplasia	Early: APC inactivation (60–80%) → β-catenin accumulation, Wnt pathway activation; Later: KRAS, SMAD4, PTEN, PIK3CA, TP53 mutations; neo-angiogenesis and stromal activation	Most common precursor; driver of sporadic CRC and FAP (multiple adenomas); progression risk increases with size, dysplasia, and villous component
Sessile Serrated Lesions (SSLs)/Serrated pathway	Mucin-vacuolated hyperplastic polyps and SSLs; serrated/“saw-tooth” crypts	Early: BRAF mutations (≥70% of SSLs), MAPK activation; Hypermethylation → CIMP, MSI; Later: silencing of tumor suppressors (TP53, CDKN2A); KRAS mutations rare (<10%), mutually exclusive with BRAF	Accounts for up to 30% of sporadic CRC; precursor lesions often in proximal colon; serrated adenocarcinomas have worse prognosis than conventional adenocarcinoma
IBD-associated dysplasia (colitis-associated CRC)	Flat or raised dysplasia; often multifocal due to ‘field cancerization’	TP53 mutations: early and frequent; also KRAS, SMAD4; APC mutations uncommon; oxidative stress and microbiota dysbiosis drive clonal expansion	Risk of CRC is 2–3× higher in IBD patients; however, ≤1% of all CRC arise from IBD; cancer often synchronous/metachronous; pathogenesis differs from sporadic MSS CRC

Abbreviations: CRC, colorectal carcinoma; SSL, sessile serrated lesion; FAP, familial adenomatous polyposis; MSI, microsatellite instability; CIMP, CpG island methylator phenotype; MAPK, mitogen-activated protein kinase; MSS, microsatellite stable; IBD, idiopathic inflammatory bowel disease.

**Table 3 pathophysiology-32-00052-t003:** Histopathologic subtypes of colorectal carcinoma (CRC). Overview of histopathologic subtypes of CRC highlighting morphologic features, molecular/pathobiologic associations, and their prognostic implications.

Histologic Subtype	Morphology/Defining Features	Molecular/Pathobiologic Features	Prognosis/Clinical Notes
Adenocarcinoma NOS	Conventional malignant glands; no special features	Heterogeneous; baseline reference category	Standard comparator; variable prognosis depending on stage
Medullary carcinoma	Solid growth; sheets of cells with vesicular nuclei, prominent nucleoli, intraepithelial lymphocytes	MSI-H; can be sporadic or Lynch syndrome–associated	Relatively favorable prognosis compared to adenocarcinoma NOS
Mucinous adenocarcinoma	Extracellular mucin pools ≥50% of tumor volume	Heterogeneous; overlaps with adenocarcinoma NOS, signet-ring, and invasive stratified mucinous carcinomas	Prognosis variable; can behave more aggressively, depending on coexisting subtype
Signet-ring cell carcinoma	Intracytoplasmic mucin displacing nucleus peripherally; diffuse infiltrative pattern	Often associated with MSI-L/MSS; overlaps with mucinous subtypes	Poor prognosis, independent of stage; highly infiltrative
Micropapillary carcinoma	Small tight clusters in cleft-like spaces; inverted polarity	Reverse cell polarity with ‘inside-out’ MUC1/EMA staining; frequent lymphovascular invasion	Aggressive course; early nodal/vascular spread
Serrated adenocarcinoma	Serrated/’sawtooth’ glandular architecture; often eosinophilic cytoplasm	Associated with serrated adenomas; frequently CIMP-high, MSI variable	Worse prognosis compared to adenocarcinoma NOS
Adenoma-like carcinoma	Morphology mimicking conventional adenomas with malignant cytology	Overlaps with adenocarcinoma NOS pathways	Clinical relevance limited; prognosis similar to adenocarcinoma NOS
Adenosquamous carcinoma	Combined glandular and squamous differentiation	Variable; may show p53 mutations	Rare, aggressive; worse prognosis than adenocarcinoma NOS
Sarcomatoid/undifferentiated carcinoma	Spindle-cell or pleomorphic morphology with minimal gland formation	EMT signatures; variable molecular profiles	Highly aggressive; poor outcomes
Neuroendocrine carcinoma (small/large cell)	Small-cell: scant cytoplasm, high N:C ratio. Large-cell: organoid growth, necrosis	Expression of neuroendocrine markers (synaptophysin, chromogranin, INSM1); TP53/RB1 alterations common; high Ki-67	Poor prognosis; aggressive, often metastatic at diagnosis
Invasive stratified mucinous carcinoma	Stratified epithelium with mucin production; overlaps with mucinous subtype	Heterogeneous; may share features with MSI or KRAS-mutated tumors	Prognosis variable; emerging recognition
Stem-cell/multidirectional carcinoma	Bidirectional/multilineage differentiation (glandular, squamous, neuroendocrine elements)	Suggests stem-cell origin; lineage plastic-ity; dedifferentiation pathways	Rare, aggressive; reflects tumor heterogeneity

Abbreviations: NOS, not otherwise specified; MSI-H, microsatellite instability–high; MSI-L, microsatellite instability–low; MSS, microsatellite stable; CIMP, CpG island methylator phenotype; EMT, epithelial–mesenchymal transition; N:C, nuclear-to-cytoplasmic ratio; INSM1, insulinoma-associated protein 1.

**Table 4 pathophysiology-32-00052-t004:** Consensus molecular subtypes (CMS) of colorectal carcinoma (CRC). Summary of the four consensus molecular subtypes of CRC (CMS1–4) including key features, molecular characteristics, prevalence, prognostic patterns, and limitations.

Subtype	Key Features	Molecular Profile	Prevalence/Stage	Prognosis and Therapy	Limitations
CMS1 (Immunogenic)	Right-sided; SSL association; immune-rich TME; CIMP-high	MSI-H, BRAF mut, immune activation	~15%; stable across stages	Early stage: good; Stage IV: worst OS; potential benefit from ICIs	Overlap with other CMS; right-sided tumors not exclusive; genotype not definitive
CMS2 (Canonical)	Left-sided; epithelial morphology; CIN phenotype	High SCNA; WNT, MYC, SRC/EGFR activation	~35%; stable across stages	Most favorable OS; best response to adjuvant chemo	Mixed molecular features common
CMS3 (Metabolic)	Mixed sidedness; epithelial; metabolic dysregulation	Low SCNA/CIMP; mixed MSI; KRAS mut; metabolic pathway activation	~15%; can transition to CMS4	Intermediate OS; adjuvant chemo benefit (like CMS2)	Not exclusive; overlap with CMS1/4
CMS4 (Mesenchymal)	Stromal-rich; CAF infiltration; EMT; angiogenesis; tumor budding	EMT, TGF-β, angiogenesis, YAP/TAZ	~25%; 10% in Stage I → 40% in Stage IV	Worst OS; poor therapy response; high metastatic risk	Therapy can shift profile (e.g., WNT resistance → YAP/TAZ reprogramming)
Mixed/Unclassified	Weighted combinations of CMS (‘continuum’ model)	Variable	~10% overall	Prognosis variable	Reflects heterogeneity and technical limits

Abbreviations: SSL, sessile serrated lesion; TME, tumor micro-environment; CIMP, CpG island methylator phenotype; MSI-H, microsatellite instability-high; mut, mutation; SCNA, somatic copy number alteration; CIN, chromosomal instability; CAF, cancer-associated fibroblast; EMT, epithelial–mesenchymal transition; OS, overall survival; ICI, immune checkpoint inhibitor.

## Data Availability

The raw data supporting the conclusions of this article will be made available by the authors on request.

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
