# Peer review of "Embryonic Signaling Pathways Shape Colorectal Cancer Subtypes: Linking Gut Development to Tumor Biology"

_pathophysiology, 2025, doi:10.3390/pathophysiology32040052_

Round 1
Reviewer 1 Report
Comments and Suggestions for Authors
This brief and structured review examines precursor lesions and Consensus Molecular Subtypes (CMSs) of colorectal cancer (CRC), with a focus on embryology and the tumor microenvironment (TME) in CRC biology. To enhance the significance of this review, a discussion of the clinical implications is essential.
This work makes a commendable attempt to summarize advances in the understanding of precursor lesions and consensus molecular subtypes of colorectal cancer (CRC), with a particular focus on embryology and the tumor microenvironment in CRC biology. While the structure is clear, the overall quality falls short of what is expected for a review article, as it reads more like lecture notes.
Abstract: The quoted statement, “The question is to what extent the embryologic development of the mid- and hindgut contributes to differences in the underlying tumor biology,” suggests that the review will focus on the relationship between embryologic development and tumor biology. However, the abstract lacks clear logical flow and appears to present multiple ideas without coherent organization.
Section 1 (Introduction): The objective of the review is not clearly stated. It would be helpful to specify what gaps or recent advances in understanding the heterogeneous nature of CRC biology remain not well reviewed by existing literature.
Sections 2 and 3: The authors devote considerable effort to discussing right- versus left-sided cancers and epithelial cancers of the midgut, but this focus seems somewhat tangential and delays engagement with the core topics of the review.
Sections 4–7: To improve clarity, it would be beneficial to include a summary table consolidating key points from these sections. Additionally, the reference list (62 citations) appears limited given the scope of the topic; a more comprehensive literature search is advised to broaden coverage.
Incorporating these suggestions will significantly strengthen the review by providing a more thorough and cohesive overview of advances in understanding precursor lesions and consensus molecular subtypes of colorectal cancer.
Author Response
We would like to thank the reviewers for taking time to review our manuscript (MS); it's much appreciated. We have revised the MS in line with their recommendations as highlighted in the 'marked-up' copy. As per the Journal's instructions, we are hereby providing a point-by-point response to the comments.
Comments 1:This brief and structured review examines precursor lesions and Consensus Molecular Subtypes (CMSs) of colorectal cancer (CRC), with a focus on embryology and the tumor microenvironment (TME) in CRC biology. To enhance the significance of this review, a discussion of the clinical implications is essential.
Response 1: The differences between right- and left CRC have clinical implications as does the CMS classification. We have now also added substantial areas that have a bearing on clinical implications, under the section marked as '8. Tumor Microenvironment'.
Comments 2: This work makes a commendable attempt to summarize advances in the understanding of precursor lesions and consensus molecular subtypes of colorectal cancer (CRC), with a particular focus on embryology and the tumor microenvironment in CRC biology. While the structure is clear, the overall quality falls short of what is expected for a review article, as it reads more like lecture notes.
 Response 2: In addition to the new section 8, the revised MS has been expanded to include 4 tables which supplement the text and improve readability.   
Comments 3: Abstract: The quoted statement, “The question is to what extent the embryologic development of the mid- and hindgut contributes to differences in the underlying tumor biology,” suggests that the review will focus on the relationship between embryologic development and tumor biology. However, the abstract lacks clear logical flow and appears to present multiple ideas without coherent organization.  
Response 3: The message in the review is: The embryology of the gut is complex but well-coordinated by signaling pathways. The same pathways are reactivated/upregulated in CRC but in an uncoordinated manner due to mutations. However, the mutations don't explain everything. The microenvironment is also important as is illustrated by the stark contrast in the incidence and repertoire of tumors from the jejunoileum, appendix and right colon (all derived from the midgut). The contrasts are further extended to the differences between the right- and left colon and rectum, again highlighting the importance of the microenvironment.
Comments 4: Section 1 (Introduction): The objective of the review is not clearly stated. The objective of the review has now been incorporated into the introduction: …The heterogeneity of colorectal cancer is partly attributable to embryologic origin and tumor location, with important distinctions between right- and left-sided tumors recognized as early as 1990 by Bufill [6, 7, 8]….
Response 4: Since then, there have been major developments in CRC biology including genomics, transcriptomics and consensus molecular subtypes. This review takes these developments into account, comparing similarities between embryonic signaling pathways which are coordinated and physiologic vis-à-vis CRC which is characterized by aberrant and uncontrolled growth due to genomic alterations. Since CRC patients on systemic therapy commonly develop drug resistance, the review also describes the limitations of the CMS classification and how studies are transitioning to focus on the tumor microenvironment.
Comments 5: It would be helpful to specify what gaps or recent advances in understanding the heterogeneous nature of CRC biology remain not well reviewed by existing literature.
Response 5:  The introduction addresses why its important to study CRC based on its prevalence in relation to other major cancers. It points out that a major problem with CRC is disease heterogeneity, which makes treatment a challenge.  
As for the gaps or recent advances; these are discussed as follows: i) The review shows the differences in the precursor lesions of CRC and how this is related to differences in the signaling pathways &/or the sequence of genomic events. ii) The growing realization of the limitations of the CMS classification is a relatively new development. This has shifted attention to the TME and how it can be manipulated to enhance therapeutic approaches (section 8).
Comments 6: Sections 2 and 3: The authors devote considerable effort to discussing right- versus left-sided cancers and epithelial cancers of the midgut, but this focus seems somewhat tangential and delays engagement with the core topics of the review.  
Response 6: A major problem in treating CRC is disease heterogeneity; one can't take a one-size-fits-all approach. To better understand the heterogeneity, it is important to look at the various topographic sites, starting with the embryologic perspective and how this has a bearing on the microenvironment. This discussion integrates the interplay between tumor site, biological behavior, and clinical implications.
Comments 7: Sections 4–7: To improve clarity, it would be beneficial to include a summary table consolidating key points from these sections.
Response 7: We have now included 4 tables to supplement the text.
Comments 8: Additionally, the reference list (62 citations) appears limited given the scope of the topic; a more comprehensive literature search is advised to broaden coverage.  
Response 8: We have added references particularly for the TME section. There are now 75 references.
 Comments 9: Incorporating these suggestions will significantly strengthen the review by providing a more thorough and cohesive overview of advances in understanding precursor lesions and consensus molecular subtypes of colorectal cancer.
Response 9: We completely agree. The suggestions have significantly improved the manuscript.
Reviewer 2 Report
Comments and Suggestions for Authors
Dear authors:
Congratulations on the excellent review you've conducted on the topic. I believe it's the tip of an iceberg that is just beginning to be seen.
Perhaps more concise outlines would be needed, since the information provided is extensive.
A chain of events or hypothesis that guides the topic would be missed.
A question that perhaps isn't very clear: What clinical implications might this type of study have?
Congratulations again.
Best regards.
Author Response
We would like to thank the reviewers for taking time to review our manuscript (MS); it's much appreciated. We have revised the MS in line with their recommendations as highlighted in the 'marked-up' copy. As per the Journal's instructions, we are hereby providing a point-by-point response to the comments.
Comments 1: Congratulations on the excellent review you've conducted on the topic. I believe it's the tip of an iceberg that is just beginning to be seen.
Response 1: We agree. The studies on tumor microenvironment potentially have wide ramifications.
Comments 2: Perhaps more concise outlines would be needed, since the information provided is extensive.
Response 2: A concise outline would be: 'One of the major limitations in the management of CRC is disease heterogeneity. It is important to understand the basis of the heterogeneity from an embryologic as well as a tumor microenvironment perspective. This helps to formulate a personalised approach to CRC management'.
Comments 3: A chain of events or hypothesis that guides the topic would be missed.
Response 3: We have also added 4 tables to the text which provides a synopsis of key features/chain of events.
Comments 4: A question that perhaps isn't very clear: What clinical implications might this type of study have?
Response 4: We have added a section (Tumor microenvironment) that includes important points about clinical implications, especially the very problematic issue of drug resistance.
Reviewer 3 Report
Comments and Suggestions for Authors
Toews et al. provided a review highlighting the role of the tumor microenvironment and embryogenesis in colorectal cancer biology. Although the subject is of interest, I believe the authors did not achieve their objective.
- First, the structure of the text must be improved. A clearer order is required to present the various topics discussed in the review in a more logical way.
- An initial suggestion would be to move the current item 4 to item 2, since embryonic development is, after all, the main focus of the manuscript.
- There should be a specific section on the tumor microenvironment in the context of CRC. Although this is a relevant topic, as indicated by the authors themselves, little or almost nothing is discussed about it (perhaps only in item 3)
- There was no clear association, or even speculation, between embryological development/tumor microenvironment and several of the topics presented, particularly regarding the classification of colorectal tumors. In some sections, the authors merely limit themselves to stating that embryogenesis may provide clues...
- A minor observation: throughout the text, many genes were not written in italics.
Author Response
We would like to thank the reviewers for taking time to review our manuscript (MS); it's much appreciated. We have revised the MS in line with their recommendations as highlighted in the 'marked-up' copy. As per the Journal's instructions, we are hereby providing a point-by-point response to the comments.
Toews et al. provided a review highlighting the role of the tumor microenvironment and embryogenesis in colorectal cancer biology. Although the subject is of interest, I believe the authors did not achieve their objective.
Comment 1: First, the structure of the text must be improved. A clearer order is required to present the various topics discussed in the review in a more logical way.
Response 1: We have now moved item #4 to #2 which better reflects the chronology. The references have been re-numbered accordingly.
Comment 2: An initial suggestion would be to move the current item 4 to item 2, since embryonic development is, after all, the main focus of the manuscript.
Response 2: We agree, and this has now been done.
Comment 3: There should be a specific section on the tumor microenvironment in the context of CRC. Although this is a relevant topic, as indicated by the authors themselves, little or almost nothing is discussed about it (perhaps only in item 3).
Response 3: We agree, the tumor microenvironment (TME) is very important. A new subsection on TME has been added (please see 'marked-up' copy.
Comment 4: There was no clear association, or even speculation, between embryological development/tumor microenvironment and several of the topics presented, particularly regarding the classification of colorectal tumors. In some sections, the authors merely limit themselves to stating that embryogenesis may provide clues...
Response 4: In the conclusion, we are saying that the same signaling pathways that guide embryologic development are upregulated/reactivated in CRC. The difference is that the process is controlled and coordinated in embryologic development whereas it is aberrant and dysregulated in cancer. The new section on TME emphasizes the importance of disease biology, particularly in relation to drug resistance and treatment failure. CRC research impetus appears to be shifting from CMSs to TME.
Comment 5: A minor observation: throughout the text, many genes were not written in italics.
Response 5: We have corrected this.
Round 2
Reviewer 1 Report
Comments and Suggestions for Authors
All comments have been addressed.
Author Response
Response: Thank you for your time.
Reviewer 3 Report
Comments and Suggestions for Authors
In my opinion, the manuscript was really improved and deserves publication. Thanks to the authors.
Author Response
Response: Thank you for your time.